# COVID-19-Associated Cardiovascular Complications

**DOI:** 10.3390/diseases9030047

**Published:** 2021-06-29

**Authors:** Clement C. E. Lee, Kashan Ali, David Connell, Ify R. Mordi, Jacob George, Elizabeth MSL Lang, Chim C. Lang

**Affiliations:** Division of Molecular & Clinical Medicine, School of Medicine, Ninewells Hospital & Medical School, University of Dundee, Dundee DD1 9SY, UK; ceclee@dundee.ac.uk (C.C.E.L.); kali001@dundee.ac.uk (K.A.); david.connell@nhs.scot (D.C.); i.mordi@dundee.ac.uk (I.R.M.); j.george@dundee.ac.uk (J.G.); elizxbethlxng@gmail.com (E.M.L.)

**Keywords:** COVID-19, SARS-CoV-2, Long COVID, cardiovascular system

## Abstract

Coronavirus disease 2019 (COVID-19) has been reported to cause cardiovascular complications such as myocardial injury, thromboembolic events, arrhythmia, and heart failure. Multiple mechanisms—some overlapping, notably the role of inflammation and IL-6—potentially underlie these complications. The reported cardiac injury may be a result of direct viral invasion of cardiomyocytes with consequent unopposed effects of angiotensin II, increased metabolic demand, immune activation, or microvascular dysfunction. Thromboembolic events have been widely reported in both the venous and arterial systems that have attracted intense interest in the underlying mechanisms. These could potentially be due to endothelial dysfunction secondary to direct viral invasion or inflammation. Additionally, thromboembolic events may also be a consequence of an attempt by the immune system to contain the infection through immunothrombosis and neutrophil extracellular traps. Cardiac arrhythmias have also been reported with a wide range of implicated contributory factors, ranging from direct viral myocardial injury, as well as other factors, including at-risk individuals with underlying inherited arrhythmia syndromes. Heart failure may also occur as a progression from cardiac injury, precipitation secondary to the initiation or withdrawal of certain drugs, or the accumulation of des-Arg^9^-bradykinin (DABK) with excessive induction of pro-inflammatory G protein coupled receptor B_1_ (BK1). The presenting cardiovascular symptoms include chest pain, dyspnoea, and palpitations. There is currently intense interest in vaccine-induced thrombosis and in the treatment of Long COVID since many patients who have survived COVID-19 describe persisting health problems. This review will summarise the proposed physiological mechanisms of COVID-19-associated cardiovascular complications.

## 1. Introduction

Coronavirus disease (COVID-19), previously known as 2019 novel coronavirus, is caused by an infection with severe acute respiratory syndrome coronavirus 2 (SARS-CoV-2). It was first detected in December 2019 among a cluster of patients presenting with pneumonia of unknown cause in Wuhan, Hubei Province, China [1]. On 11 March 2020, it was declared a pandemic by the World Health Organisation (WHO), and as of 27 December 2020, more than 79 million cases have been reported worldwide with more than 1.7 million deaths [2,3].

Like SARS-CoV and MERS-CoV, SARS-CoV-2 is from the genus Betacoronavirus from within the subfamily of *Coronavirinae,* which arises from the family of *Coronoaviridae* [1]. It is an enveloped and non-segmented, positive-sense RNA virus with a length of about 29.9kb [4] with huge genomic diversity [5].

A variety of outcomes have been observed amongst patients. In an analysis of 44,672 COVID-19-positive patients in China, 81% were mild (no or mild pneumonia), 4.7% were critical (respiratory failure, septic shock, and/or multiple organ dysfunction/failure), and mortality rate was 2.3% [6]. Patients requiring ICU care were also significantly older (median age, 66 years vs. 51 years) than those who did not [7]. SARS-CoV-2 infection has been associated with cardiovascular complications (Figure 1), namely myocardial injury and myocarditis (36% of 2736 patients), thromboembolic events (25% of 81 patients), heart failure and cardiomyopathy (23% of 191 patients), arrhythmias (16.7% of 138 patients), and acute coronary syndromes (0.96% of 4702 patients) [7,8,9,10,11,12]. Furthermore, COVID-19 symptoms and complications may persist for months after the initial infection in what has been termed as Long COVID.

## 2. Epidemiology of Cardiovascular Complications

Thromboembolic events such as deep vein thrombosis (DVT), pulmonary embolism (PE), and arterial thrombosis have been observed frequently during the infection [13,14]. Among patients in the intensive care unit (ICU), as many as 25% developed venous thromboembolisms, and the incidence of PEs were found to be more common in COVID-19 acute respiratory distress syndrome (ARDS) as compared to ARDS of other causes [9,15,16]. Furthermore, an autopsy sample of 12 patients who had died of COVID-19 had an incidental finding of DVT, and a third were discovered to have had a PE as the cause of death [17]. The pooled mortality of arterial and venous thromboembolisms, based on a systemic review and meta-analysis of 8271 patients across UK, USA, Europe, and China, was estimated to be at 23% [18]. 

Myocardial injury was another complication commonly reported in patients with acute COVID-19 infection. It is defined as an evidence of elevated cardiac troponin levels with at least one value above the 99th percentile upper reference limit in the absence of myocardial ischaemia [19]. Presenting features include chest pain, dyspnoea, dysrhythmia, and acute left ventricular dysfunction. A study of 2736 SARS-CoV-2 patients in the U.S. found that up to 36% have elevated troponin levels early in the course of the disease [10,20]. Those with pre-existing coronary artery disease, cerebrovascular disease, and chronic HF were more likely to develop myocardial injury [10,21]. Elevated troponins during hospitalisation were also associated with an increased risk of requiring ventilation, fatal ventricular arrhythmias, and a 59.6% mortality risk [21,22].

While uncommon, there have been reports from the U.S. and Italy of acute coronary syndromes (ACS) occurring as the initial presentation or during hospitalisation for COVID-19 [23,24]. Among these cases, approximately one-third were found to have non-obstructive coronary artery disease [23,24]. A multicentre study consisting of 4702 patients across 17 sites in Italy, Spain, and Switzerland observed ACS to occur in 0.96% of COVID-19 patients, with an in-hospital mortality rate of 27.3%. This observed mortality rate was thrice that of SARS-CoV-2-negative patients presenting with ACS [11].

Arrhythmia was a common complication observed in 16.7% of 138 SARS-CoV-2-positive patients in a hospital in China [7]. It was observed to occur more frequently in patients requiring ICU admission than those who did not [25]. The development of acute malignant arrhythmias, such as ventricular tachycardia, ventricular fibrillation, or atrioventricular block in patients hospitalised for COVID-19, was associated with a significantly increased mortality [26]. In a systematic review and meta-analysis of 17435 patients across 11 countries, mortality was found to be 20.3% amongst hospitalised COVID-19 patients who developed arrhythmia [27]. Higher levels of troponin have been correlated with an increased incidence of malignant ventricular arrhythmia, suggesting that arrhythmia may be the consequence of other complications [22,25,26].

Heart failure (HF) was observed in 23% of 191 SARS-Cov-2-positive inpatients in China [12]. A retrospective study of 113 deceased patients observed that nearly half had HF as a complication [28]. Furthermore, those with underlying cardiovascular co-morbidities, such as chronic hypertension, were more likely to develop HF [28]. HF patients were found to be at a higher risk of thromboembolic events, ARDS, severe hypotension, and death [29]. The mortality rate of 77 SARS-CoV-2-positive patients in a hospital in Spain who developed acute HF was observed to be as high as 46.8% [30].

Signs and symptoms of cardiovascular complications, such as dyspnoea and chest pain, may have significant overlap with COVID-19. In addition, cardiovascular complications have been observed to occur frequently during the course of the disease. As such, cardiovascular complications should always be considered especially in more severe COVID-19 cases, such as those requiring hospitalisation. A group of patients that may have to be monitored more carefully are those with underlying cardiovascular co-morbidities. It was observed that these patients were more likely to require hospitalisation, admission to ICU, and had poorer mortality outcomes [6,7,31,32,33].

The diverse and frequent cardiovascular complications observed in a respiratory infection such as COVID-19, as well as the preponderance for those with cardiovascular co-morbidities, suggests the complex nature of this disease. We will now review the potential pathophysiological mechanisms that underlie these cardiovascular complications.

## 3. Pathophysiology of COVID-19-Associated Cardiovascular Complications

Multiple factors possibly contribute to the cardiovascular complications observed in association with COVID-19 infections. These may include systemic inflammation, direct invasion of cardiovascular tissue, and medications and so forth. (Figure 2) The following sections will review the potential mechanisms as to how cardiovascular complications may arise at various stages of the disease.

### 3.1. SARS-CoV-2 Entry into Cells

Viruses such as SARS-CoV-1 and SARS-CoV-2 have been known to interact with ACE2 [34,35]. Complications can arise as a direct consequence of the virus’ entry into tissue that express ACE2. ACE2 is a widely expressed, membrane-associated aminopeptidase found in the heart, lungs, pericytes, and vessels, among others [1,25,36]. The transmembrane spike glycoprotein that protrudes from SARS-CoV-2’s surface allows interaction with ACE2 and facilitates the virus’s entry into the cell [37]. Entry into myocardial tissue could disable or destroy the host cell, contributing to the release of danger signals and activation of the host’s immune response, resulting in myocardial injury and myocarditis [20,25,38]. Entry of the virus into pericytes may potentially cause local microvascular inflammation, leading to severe microvascular dysfunction and, consequently, myocardial infarction [1]. The combination of the damage caused to myocardial or conducting tissue and a SARS-CoV-2-induced ACS may contribute to arrhythmia [25]. Moreover, HF patients express high levels of ACE2, potentially increasing the risk of a direct viral infection of the myocytes and further exacerbation [39]. Direct invasion of the vascular endothelium via ACE2 receptors may contribute to disordered cytokine signalling alongside the release of damage-associated molecular patterns (DAMPs), which may contribute to thrombi formation [40]. Viral entry via ACE2 receptors present on the pulmonary endothelial surface may result in the formation of pulmonary microthrombi as a means to limit viral invasion through a process known as immunothrombosis [41]. While the cardiovascular complications may be a result of the site of viral invasion, ACE2 also interacts with other molecular pathways that can potentially affect the cardiovascular system.

Similar to SARS-CoV-1, the binding of SARS-CoV-2 to ACE2 has been hypothesised to downregulate the receptor’s function of degrading angiotensin II [34]. Excessive angiotensin II potentially activates the pro-thrombotic p38 Mitogen-Activated Protein Kinase (MAPK) pathway, which may lead to thromboembolic events [35]. Angiotensin II may also induce Nox2-related reactive oxygen species (ROS) production, resulting in myocardial injury [42]. Downregulation of ACE2 may cause des-Arg^9^-bradykinin (DABK) accumulation and excessive G-protein-coupled receptor B_1_ (BK_1_) activation, potentially contributing to HF [43]. Beyond the local inflammatory effects caused by the virus’s entry, a systemic inflammatory response has been proposed to play an essential role in cardiovascular complications.

### 3.2. The Role of Systemic Inflammation

It is hypothesised that SARS-CoV-2 may induce an excessive acute systemic inflammatory response causing endothelial dysfunction and the activation of complement pathways, platelets, von Willebrand factor, Toll-like receptors, and tissue factor pathway. This may result in venous and arterial thrombosis [15,44,45,46]. Endothelial dysfunction, together with pathogenic complement activation, may result in diffuse thrombotic microangiopathy, further contributing to thrombosis [14]. The infection has also been observed to induce a disproportionate increase in factor VIII, high concentrations of circulating microvesicles, and neutrophil extracellular traps (NETs), all components that may increase coagulability [15,47]. Furthermore, NETs which are pro-inflammatory and pro-thrombotic have been found to be elevated in SARS-CoV-2-positive patients, which could be an explanation for the thrombosis observed in COVID-19 patients, including in the coronary circulation [47,48,49]. Apart from thrombotic events, systemic inflammation may also contribute to other cardiovascular complications.

The systemic inflammation secondary to the infection may result in an increase in metabolic demand, which may be unmet by cardiomyocytes, potentially precipitating an ACS [10,50]. Virally-induced systemic inflammation may also precipitate coronary plaque ruptures and also increases the likelihood of stent thrombosis [50]. A T-helper-1 (Th1) mediated cytokine storm may potentially contribute to myocardial injury [31,51]. In addition, inflammatory cytokines can potentially induce cardiac sympathetic hyperactivation via hypothalamus-mediated and peripheral pathways [52]. Specific cytokines, such as interleukin (IL)-6, derived from T cells and macrophages, have been implicated in the inflammatory response and will be further elaborated [53,54].

### 3.3. The Role of IL-6

IL-6 has multiple potential interactions that have been thought to play an essential role in cardiovascular complications. IL-6 levels were elevated in COVID-19 patients and could potentially interact with NETs to drive thrombosis [55]. This pro-inflammatory cytokine has been thought to induce T lymphocyte activation, resulting in myocardial injury [56]. Together with other pro-inflammatory cytokines, such as IL-1 and tumour necrosis factor-α, they have been shown to modulate the expression and/or function of K^+^ and Ca^2+^ ion channels in cardiomyocytes, potentially causing arrhythmias [25,52]. IL-6 may potentially inhibit human Ether-a-go-go-Related Gene (hERG) K^+^ channels, resulting in QT interval prolongation [52,57]. It may also inhibit CYP3A4, thus increasing the bioavailability of certain QT-prolonging drugs [52]. In addition, IL-6 may displace plakoglobin, a desmosomal protein from the cardiomyocyte membrane, which can result in cardiac cell death and fibrofatty-replacement processes, which can contribute to arrhythmia [56]. IL-6 may also play a role in acute HF [58].

### 3.4. Medications and Cardiovascular Complications

Beyond the effects of a direct viral invasion and the body’s inflammatory response, medications and investigational drug therapies may contribute to cardiovascular complications. An extensive review of this subject is beyond the scope of this review. We have focused on some cardiovascular complications related to chloroquine and the anti-virals lopinavir, ritonavir and remdesivir, which have been approved for the treatment of COVID-19 (Table 1) [59]. Chloroquine and hydroxychloroquine (CQ/HCQ) are thought to alter endosomal pH and reduce glycosylation of the ACE2 receptor, thus preventing viral entry [60,61]. However, they can potentially prolong QT intervals by blocking the hERG K^+^ channel, which can result in torsade de pointes or sudden cardiac death [60]. An acute kidney injury secondary to a COVID-19 infection may accumulate chloroquine/hydroxychloroquine, resulting in further QT prolongation [62]. Concomitant use of azithromycin use with hydroxychloroquine increases the risk of QTc prolongation, especially in patients with high levels of transaminases, which likely indicate a severe inflammatory response [57]. Loop diuretics, commonly used in volume management and ARDS, have also been shown to increase the likelihood of prolonged QTc in patients taking hydroxychloroquine [63]. Chloroquine and hydroxychloroquine may also cause lysosomal dysfunction and accumulation of glycogen and phospholipids, the effects of which are cardiotoxic and can result in HF [64].

Another anti-viral therapy proposed for use in SARS-CoV-2 infections, lopinavir-ritonavir, inhibits CYP3A4, potentially decreasing serum concentrations of the active metabolites of P2Y_12_ inhibitors, such as clopidogrel, which could result in thrombosis [8]. Lopinavir-ritonavir can also inhibit CYP2D6, which potentially increases chloroquine plasma levels, resulting in malignant arrhythmias [65]. Apart from these investigational therapies, other factors, ranging from channelopathies to medication compliance, may also contribute to cardiovascular complications. As for the approved anti-viral agent, remdesivir, there has been reported episodes of bradycardia related to its use [66].

### 3.5. Other Contributing Factors

Inherited arrhythmia syndromes may be precipitated during a COVID-19 infection. Fever is known to trigger arrhythmia in the presence of LQTS 2 mutations (LQTS) or SCN5A mutations (Brugada syndrome), while diarrhoea can potentially cause hypokalaemia, affecting QTc interval and thus can be detrimental in these inherited arrhythmia syndromes [67].

Withdrawal of Renin-Angiotensin-Aldosterone system (RAAS) inhibitors due to misinformed concerns of increased infection risks can also contribute to decompensation in HF patients, especially since plasma levels of angiotensin II, aldosterone, cortisol, norepinephrine, left ventricular end-diastolic, and end-systolic volumes return to pre-treatment values upon cessation of these drugs [68,69]. While the incidence is not yet known, patients may possibly stop anti-coagulant medications due to misinformation, which can potentially result in a thrombotic event [70].

Severe illness, hypoxia, mechanical ventilation, central venous catheterisation, dehydration, obesity, and immobility may contribute to the observed thromboembolic events [70,71]. COVID-19, being an acute respiratory infection, may also trigger atrial fibrillation (AF), which could also be a potential cause of thromboembolic events [72]. While the exact incidence of new-onset AF is unknown, it has been estimated to occur in 3.6 to 6.7% of COVID-19 patients based on some case reports and clinical studies [73]. Pulmonary hypertension secondary to ARDS and/or PE may lead to HF [25]. Furthermore, electrolyte derangements and adrenergic stress can lead to electrical instability [25].

A diverse range of proposed pathophysiological mechanisms potentially underlie the cardiovascular complications associated with an ongoing COVID-19 infection. Understanding these mechanisms may, therefore, aid in devising a targeted management plan that can potentially reduce the occurrence or improve the outcomes of associated cardiovascular complications. While these complications were observed to occur during the initial phase of the infection, emerging evidence suggests that cardiovascular complications may extend beyond. This will be touched upon in the next section.

## 4. Long COVID

With the progression of the global COVID-19 pandemic, there is increasing evidence that some patients are experiencing prolonged symptoms often involving multiple organs and systems beyond the initial period of the acute COVID-19 infection. There is now guidance from the National Institute for Health and Care Excellence regarding what has been termed Long COVID [74]. Long COVID includes ongoing signs and symptoms of COVID-19 for 4 to 12 weeks after the acute infection, and/or the persistence or development of signs and symptoms beyond 12 weeks from the onset of a COVID-19 infection [75]. However, it should be noted that it has not been fully defined, as there is still much to understand about this condition [76]. Signs and symptoms include fatigue, dyspnoea, joint pain, chest pain, cough, palpitations, anosmia, dysgeusia, alopecia, cognitive blunting, and psychological distress (Table 2) [77,78]. It is of concern that after 60 days from the onset of infection, as many as 87.4% of 143 patients in an Italian study were affected by Long COVID symptoms [77,79].

The use of imaging modalities may help us to understand the cardiovascular involvement in Long COVID. Imaging studies have observed cardiovascular changes that occur following a COVID-19 infection [80,81,82,83]. Myocardial oedema and late gadolinium enhancement (LGE) have been observed in 58% of 26 Chinese patients presenting with Long COVID cardiac symptoms [84]. Myocarditis-pattern LGE have been observed up to a month post-infection in 27% of 148 patients in the UK who have had myocardial injury; among which, one-third had findings consistent with an active myocarditis [85]. Furthermore, up to 78% of 100 mild COVID-19 cases in Germany—despite having fewer co-morbidities—had CMR abnormalities of similar extent and severity to that of patients requiring hospitalization [86]. Thus, while cardiovascular changes have been observed following a COVID-19 infection, its relationship with Long COVID is still unclear. Furthermore, we are unsure if there are other cardiovascular changes that have been yet to be detected.

## 5. Vaccine-Induced Thrombosis

Cardiovascular complications are not yet out of the picture despite the introduction of vaccines against COVID-19. There have been reports streaming in on the occurrence of thromboembolic events following ChAdOx1 nCov-19 (AstraZeneca) vaccination and potentially the Ad26.COV2.S (Johnson & Johnson) vaccination. While rare, thrombosis was observed to occur at unusual sites, such as cerebral and splanchnic veins [87,88,89]. Based on the observation of thrombocytopenia and raised antibodies to platelet factor 4-polyanion complexes, it has been suggested to be an immune-mediated reaction [87,88]. This is of importance, as clinicians must be vigilant for cardiovascular complications not just in the acute or convalescent phase of a COVID-19 infection but also after a seemingly innocuous vaccination against the virus.

## 6. Conclusions

SARS-CoV-2 was first detected in China in 2019 and has spread across the globe with widespread effects. More than just a respiratory illness, it has shown a preponderance to those with underlying cardiovascular co-morbidities [31]. In addition, cardiovascular complications have occurred frequently in association with the disease and even months after the infection [8,84,85,86,92]. These cardiovascular complications include myocardial injury and myocarditis, acute coronary syndromes, heart failure, arrythmias, and thromboembolic events. Some of the medications utilised to treat COVID-19 also have potential cardiac complications [8,57,60,61,62,63,65].

In addition, cardiac symptoms, such as palpitations, chest pain, and dyspnoea and CMR changes, have been observed in patients weeks to months after the initial infection in what has been termed Long COVID [77,79,84,85,86]. Autonomic dysfunction could be a contributor to the underlying pathophysiological mechanism of these symptoms [92]. Chronic myocarditis leading to myocardial fibrosis and the subsequent development of arrhythmias may potentially account for some of these symptoms as well [56]. The significance of the observed cardiovascular changes is still not well understood; nor do we know if there will be more to be detected [76].

The exact pathophysiological mechanisms underlying the disease and its cardiovascular sequelae are still largely speculative. Further research must be performed to better understand COVID-19 and its long-term complications.

## Figures and Tables

**Figure 1 diseases-09-00047-f001:**
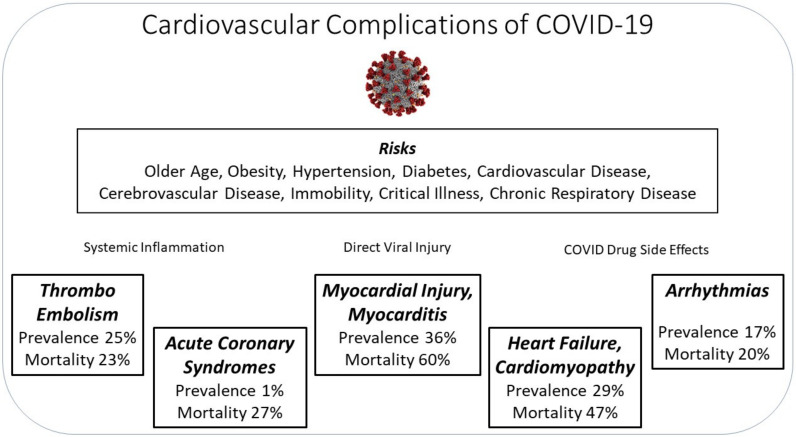
Summary of the cardiovascular complications associated with COVID-19.

**Figure 2 diseases-09-00047-f002:**
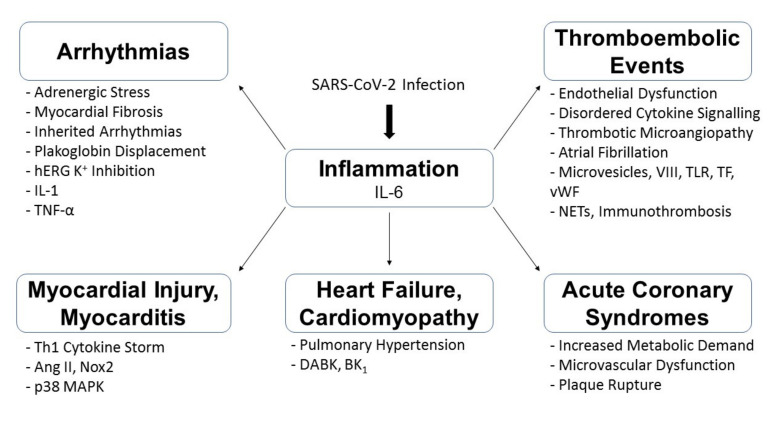
Summary of the possible pathophysiological mechanisms underlying COVID-19-associated cardiovascular complications. p38 Mitogen-Activated Protein Kinase (p38 MAPK); Angiotensin II (Ang II); T-Helper-1 (Th1); Nicotinamide Adenine Dinucleotide Phosphate Oxidase 2 (Nox2); Neutrophil Extracellular Traps (NETs); von Willebrand (vWF) factor; Toll-Like Receptor (TLR); Tissue-Factor (TF); Damage-Associated Molecular Patterns (DAMPs); Factor VIII (VIII); Human Ether-a-go-go-Related Gene (hERG); Interleukin (IL); Tumour Necrosis Factor (TNF); Des-Arg^9^-Bradykinin (DABK); G-protein coupled receptor B1 (BK_1_).

**Table 1 diseases-09-00047-t001:** Summary of proposed COVID-19 therapies and their potential cardiovascular complications [8,57,60,61,62,63,64,65,66]. Hydroxychloroquine (HCQ); Chloroquine (CQ); Acute Kidney Injury (AKI); Cytochrome P450 2D6 (CYP2D6); Cytochrome P450 3A4 (CYP3A4).

Summary of Proposed COVID-19 Therapies and Potential Cardiovascular Complications		
Proposed Therapy	Potential Cardiovascular Effects	Possible Mechanisms
HCQ or CQ	QT Prolongation	Blockage of hERG K^+^ channels
		Toxic HCQ/CQ accumulation secondary to AKI
	Cardiotoxicity	Lysosomal dysfunction → glycogen/phospholipid accumulation
HCQ + Azithromycin	QTc Prolongation (especially those with elevated transaminases)	Azithromycin prolongs QT interval Elevated transaminases indicative of pro-inflammatory cytokines which prolong QT
HCQ + Loop Diuretics	QTc Prolongation	Electrolyte derangement by loop diuretics might be a contributory factor
Lopinavir + Ritonavir	Thrombosis (in clopidogrel users)	CYP3A4 inhibition thus decreased active metabolites of clopidogrel
	Arrhythmia	CYP2D6 inhibition → CQ accumulation
Remdesivir	Bradycardia	Unclear

**Table 2 diseases-09-00047-t002:** Prevalence of Long COVID Manifestations [7,10,12,16,21,31,77,78,85,90,91]; (-) indicates that the sign/symptom has not been observed; highlighted in blue denotes the signs/symptoms common to both acute COVID-19 infection and Long COVID.

Prevalence of Cardiovascular Manifestations		
Signs/Symptoms	Acute COVID-19	Long COVID
Palpitations/Arrhythmia	16.7%	11.2%
Chest Pain	3.4%	12.3%
Myocarditis	36.0%	27.0%
Dyspnoea	31.2%	26.0%
Thromboembolism	25.0%	-
Heart Failure	23.0%	-
**Prevalence of Other Manifestations**		
Signs/Symptoms	Acute COVID-19	Long COVID
Cough	59.4%	22.0%
Myalgia/Arthalgia	34.8%	7.6%
Fatigue	69.6%	28.3%
Anosmia	41.0%	22.7%
Ageusia	38.2%	22.7%
Fever	98.6%	-
Sputum Production	26.8%	-
Headache	6.5%	-
Haemoptysis	5.0%	-
Diarrhoea	10.1%	-
Nasal Congestion	4.8%	-
Rhinorrhoea	2.4%	-
Alopecia	-	28.6%
Cognitive Blunting	-	15.0%
Psychological Distress	-	22.7%

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
