# Peer review of "COVID-19-Associated Cardiovascular Complications"

_diseases, 2021, doi:10.3390/diseases9030047_

Round 1

Reviewer 1 Report

Covid-19 disease has been associated with a number of cardiovascular complications. This review aims to summarize the mechanisms underlying these events. It is highly informative and well-structured. I only have minor comments.

  1. The introductory paragraph showing the percentages for each cardiovascular event is very helpful. I strongly suggest, however, that the authors add the total number of patients from which these percentages were extrapolated.
  2. The authors state the countries from which specific data is available. For instance, they state that arrhythmia was a common complication in Chinese patients. It would be useful to add the countries in all complications.
  3. Patients with underlying CV disease were more likely to develop HF. Was this CAD or other forms of heart disease as well?
  4. In section 3.1 the authors state that myocardial fibrosis may develop as a result of myocarditis providing a substrate for arrhythmias. Deposition of fibrous tissue, however, typically takes time. How relevant is this at the very first stage of the infection, ie. The viral entry into the cells
  5. The authors state a number of pathways that can be activated downstream of ACE2. It is not clear whether these have been shown to be dysregulated in Covid-19 disease in particular.
  6. Is the source of circulating IL-6 known? Is it over-expressed and over-secreted from cardiac myocytes themselves?
  7. I would kindly suggest for a few numbers to be added in the review. For instance, how many patients discontinued anti-coagulant medications leading to a thrombotic event? How many patients developed AF? 87.4% of patients have Long-COVID. How large was this population?

Author Response

  1. The introductory paragraph showing the percentages for each cardiovascular event is very helpful. I strongly suggest, however, that the authors add the total number of patients from which these percentages were extrapolated.
    1. We thank the reviewer for the comments. We have now added the total number of patients from which the data was derived from.
  2. The authors state the countries from which specific data is available. For instance, they state that arrhythmia was a common complication in Chinese patients. It would be useful to add the countries in all complications.
    1. We thank the reviewer for the comments. We have now included the other countries (e.g. Europe, U.S. etc) from which these complications were observed, however recognising that it would not be possible for this review to cover the complications reported from all countries.
  3. Patients with underlying CV disease were more likely to develop HF. Was this CAD or other forms of heart disease as well?
    1. We thank the reviewer for pointing this out. We have now changed this to reflect better the paper that was referenced. Rather than CAD, it was patients with cardiovascular co-morbidities such as chronic hypertension and diabetes.
  4. In section 3.1 the authors state that myocardial fibrosis may develop as a result of myocarditis providing a substrate for arrhythmias. Deposition of fibrous tissue, however, typically takes time. How relevant is this at the very first stage of the infection, ie. The viral entry into the cells
    1. We thank the reviewer for the comments. We agree that processes such as interstitial or replacement fibrosis do take time to develop. We have placed this in the section ‘Long-COVID’ as one of the possible explanations of arrhythmia in the convalescent period
  5. The authors state a number of pathways that can be activated downstream of ACE2. It is not clear whether these have been shown to be dysregulated in Covid-19 disease in particular.
    1. We thank the reviewer for the comments. We have now added that other viruses, such as SARS-CoV are also known to enter cells via the ACE2 receptor. We have also made it clearer that the mentioned dysregulation of ACE2 is still a hypothesis.
  6. Is the source of circulating IL-6 known? Is it over-expressed and over-secreted from cardiac myocytes themselves?
    1. We thank the reviewer for the comments. The source of IL-6 was not explicitly mentioned. We have now added that this could be released from T cells or macrophages as part of the general inflammatory response.
  7. I would kindly suggest for a few numbers to be added in the review. For instance, how many patients discontinued anti-coagulant medications leading to a thrombotic event? How many patients developed AF? 87.4% of patients have Long-COVID. How large was this population?
    1. We thank the reviewer for pointing this out. We are unsure of the number of patients who had discontinued anti-coagulant medications. This has been updated to state clearly that it was a theoretical risk, as CDC had in the earlier stages of the pandemic warned against ‘blood thinners’ in a document which has now been removed (https://www.cdc.gov/coronavirus/2019-ncov/downloads/community-mitigation-strategy.pdf) We have now also included the incidence of new-onset AF based some case reports and small clinical studies. The population size of the 87.4% has now also been included.

Reviewer 2 Report

This manuscript aims to review the COVID-19 associated cardiovascular complications. In general, it is well organized and easy to read.

In the Introduction, the authors gave a nice summary regarding the CV complications in Fig. 1. The possible pathophysiological mechanisms were illustrated in Fig. 2. and the signal molecules responsible for the down-regulation of Ang-II degradation were elaborated. 

However, regarding the possible adverse effects related to medications for COVID-19, it would be nice if there is a summary table or illustration. For the comparison between acute vs. long COVID-19, the incidence of each symptom/sign, rather than + or -, will be more helpful. 

Author Response

We thank the reviewer for the comments.

  • We agree with the reviewer and have now included a table to summarise the possible adverse effects related to the usage of the proposed COVID-19 therapies.
  • While the incidence of the Long-COVID signs/symptoms are not very well known, we have added the prevalence of each sign/symptom for a better contrast between Acute & Long-COVID. 

Round 2

Reviewer 2 Report

The revised manuscript is good enough to be published. 

Minor critics: 

Table I: 

"Prevalence of Cardiovascular Manifestations."--> Please remove the "."